

# Effects of harvest stages and lactic acid bacteria additives on the nutritional quality of silage derived from triticale, rye, and oat on the Qinghai-Tibet Plateau

Jun Ma, Hanling Dai, Hancheng Liu and Wenhua Du

College of Pratacultural Science, Gansu Agricultural University, Lanzhou, Gansu, China

## ABSTRACT

**Background:** Triticale (×*Triticosecale* Wittmack L.), rye (*Secale cereale* L.), and oat (*Avena sativa* L.) are the main forage crops on the Qinghai–Tibet Plateau, but there has been relatively little research on the silage produced from these three species.
**Methods:** Plants were harvested at the heading, flowering, grouting, milky, and dough stages and then used to produce silage with and without additives (Sila-Max and Sila-Mix). The nutritional quality of the resulting silages was analyzed.
**Results:** Triticale was revealed to be more suitable than oat or rye for producing silage on the Qinghai–Tibet Plateau. On the basis of the dry matter yield (DMY), triticale and rye should be harvested at the milky stage to optimize silage quality, whereas oat should be harvested at the dough stage. The lactic acid bacteria additives Sila-Max and Sila-Mix had no significant effect on the nutritional quality of the three silages regardless of when the samples were harvested. Overall, triticale produced higher quality silage than oat or rye. More specifically, triticale variety 'Gannong No.2' harvested at the milky stage is ideal for silage production.

# INTRODUCTION

The Qinghai–Tibet Plateau has a cold climate that limits the production of the relatively few forage species in the region, especially in winter (*Yuan et al., 2012*). The insufficient production of feed causes several problems for livestock, including weight loss, low milk production, low reproductive performance, and health-related issues (*Zhang et al., 2015*). Farmers in the area typically use hay as reserve forage to feed their livestock in winter. However, frequent rainfall and low temperatures during the forage harvest period in this region adversely affect hay production. Thus, it is very difficult to produce high-quality hay (*Zhao et al., 2018*). Accordingly, farmers in this region have focused on silage production to solve the problems associated with poor forage production (*Chen et al., 2016*). Compared with hay, the production of forage silage may be better for minimizing the loss of valuable nutrients (*Chen et al., 2020*; *Ribeiro et al., 2014*). Forage silage quality has commonly been increased *via* the use of additives (*e.g.*, lactic acid bacteria additives) (*Sun et al., 2012*; *Puntillo et al., 2020*). More specifically, the use of lactic acid bacteria additives

Corresponding author
Wenhua Du, duwh@gsau.edu.cn

during silage production increases feed digestibility and fermentation quality (*Gao et al., 2019*; *Petrova et al., 2022*), while also enhancing silage palatability (*Przybyło et al., 2020*). The addition of lactic acid bacteria reportedly increases the fermentation quality of maize silage by decreasing the pH and the content of neutral detergent fiber (NDF) and acid detergent fiber (ADF), minimizing the loss of dry matter (DM), and inhibiting protein degradation (*Zuo et al., 2020*). Feeding livestock with silage fermented using lactic acid bacteria increases DM intake and digestion, body weight, and milk production (*Nair et al., 2020*; *Zhang et al., 2020*).

On the Qinghai–Tibet Plateau, oat is a common forage crop that has been cultivated for a long time (*Jing et al., 2017*). Rye has recently been cultivated by local farmers because of the limited forage production in the region (*Wang, Tian & Du, 2021*). Triticale has also been grown in this region because of its high yield and quality as well as its lodging and disease resistance (*Jun et al., 2022*). Recent research indicated that triticale and oat have a DM yield (DMY) of 16.26 and 15.8 t/ha, respectively, and a crude protein (CP) content of 11.08% and 10.7%, respectively (*Wang et al., 2020*; *Zhao et al., 2018*). In terms of the rye cultivated on the Qinghai–Tibet Plateau, it has a hay yield of 10.87 t/ha, a CP content of 10.18%, and a DM digestibility (DMD) of 58.83% (*Zhao et al., 2020*). These forage species cultivated locally may provide sufficient materials for silage production.

The DMY and nutritional quality of forage species vary among growth stages. Because of the delayed growth from the booting stage to the soft dough stage, the DM, NDF, ADF, starch, and nonfiber carbohydrates contents of forage gradually increase, whereas CP content decrease (*Lyons et al., 2019*). Therefore, the nutrient content of the substrate provided by forage for silage fermentation at different harvest stages is different, and the fermentation quality and nutritional quality of silage produced at different harvest periods are also different. Thus, there is some controversy regarding the appropriate stage for harvesting triticale, rye, and oat used to produce silage. *Yang et al. (2013)* reported that the nutritional value of oat silage is higher than that of hay, and CP of oat silage that oat harvested at the heading stage more lost compared to the milky stage, so the optimal stage for harvesting oat in Yangling, Xian, China is the milky stage. Oat that will be used to produce silage should ideally be harvested at the dough stage on the Qinghai–Tibet Plateau (*Zhao et al., 2018*). For the production of high-quality silage, triticale grown in Temuco, Chile should be harvested at the full flowering stage (*Rojas et al., 2004*). In contrast, *Jung et al. (2021)* suggested that triticale cultivated in Jangsoo, Chunbuk, South Korea should be harvested at the heading stage to optimize silage quality. We hypothesised that: (1) delaying harvest stage would result in higher DMY, but it had an impact on the nutritional quality of silage; (2) lactic acid bacteria additives at all harvest stages would improve the nutritional quality of silage compared with the control; (3) the nutritional quality of different forage silage was different at the same harvest stage. Thus, we analyzed the five main growth stages of three forage species cultivated on the Qinghai–Tibet Plateau to determine when the crops should be harvested to maximize silage quality. Relatively few studies have investigated the optimal harvest time and lactic acid bacteria additive for preparing high-quality silage from triticale, rye, and oat grown on the Qinghai–Tibet Plateau. The objective of this study was to clarify the effects of harvest stages and additives
on the nutritional quality of triticale, rye, and oat silages and determine the optimal forage species, harvest stage, and lactic acid bacteria additive for producing high-quality silage on the Qinghai–Tibet Plateau.

## MATERIALS AND METHODS

### Silage preparation

The experimental work was conducted at the experimental field station (34°55′N, 102°53′E, altitude 2,950 m) in Hezuo City, Gansu, China during 2017–2018. The mean annual temperature at the study site was 2.7 °C, and the mean annual rainfall of this area was 550–680 mm. The rainfall and air temperature during growing seasons in 2018 were showed in Fig. 1. The study site contained subalpine meadow soil with a pH of 7.4. Additionally, the soil (0–20 cm depth) contained organic matter of 13.9%, available nitrogen of 248.0 mg·kg$^{-1}$, available phosphorus of 5.0 mg·kg$^{-1}$, and available potassium of 198.0 mg·kg$^{-1}$.

### Experimental materials

'Gannong No.2' triticale, 'Gannong No.1' rye, and 'Minxian' oat which was the commonly cultivated local oat variety were used in this study. The lactic acid bacteria additives were Sila-Max and Sila-Mix (Ralco Nutrition Co., Marshall, MN, USA). The composition of Sila-Max and Sila-Mix is described in the Supplemental File.

### Experimental methods

A split-plot design was used in this study, and the main plots contained the following three forage species: triticale, rye and oat which were referred to as A1, A2 and A3, respectively. Subplots were used to analyze the effects of the following harvest stages: heading, flowering, grouting, milky, and dough, which were referred to as B1, B2, B3, B4 and B5, respectively. The harvest stages of forages were showed in Table 1.

The following sub-subplots were used to evaluate the effects of additives: additive-free treatment, Sila-Max, and Sila-Mix, which were referred to as C0, C1 and C2. A total of 135 plots (4 m$^2$ each; 1 m × 4 m) were prepared. Seeds were sowed in line, with a spacing of 20 cm, a depth of 3–4 cm, and a seeding rate of 7.5 × 10$^6$ per hectare. Prior to sowing, diammonium hydrogen phosphate was applied at a rate of 300 kg·ha$^{-1}$ and also urea was applied by top-dressing (225 kg·ha$^{-1}$) at the jointing stage. In September 2017, triticale was sown, while in April 2018, rye and oat were sown, respectively.

Triticale, rye and oat were harvested at five stages, respectively, after which silage was produced immediately. During harvest, the fresh weight of each plot was determined by electronic balance (model XK3190-A32E; Xiangchuan Electronic Weighing Instrument Co., Shanghai, China). The DM content was measured using the method described by Na et al. (2013), and the dry matter yield of each plot was determined according to DM content of 500 g sample of each plot. After harvesting, all fresh plants chopped directly into 2 cm segments with a fodder chopper (model 600; Yamao Mechanical Equipment Co., Zhengzhou, Henan, China).

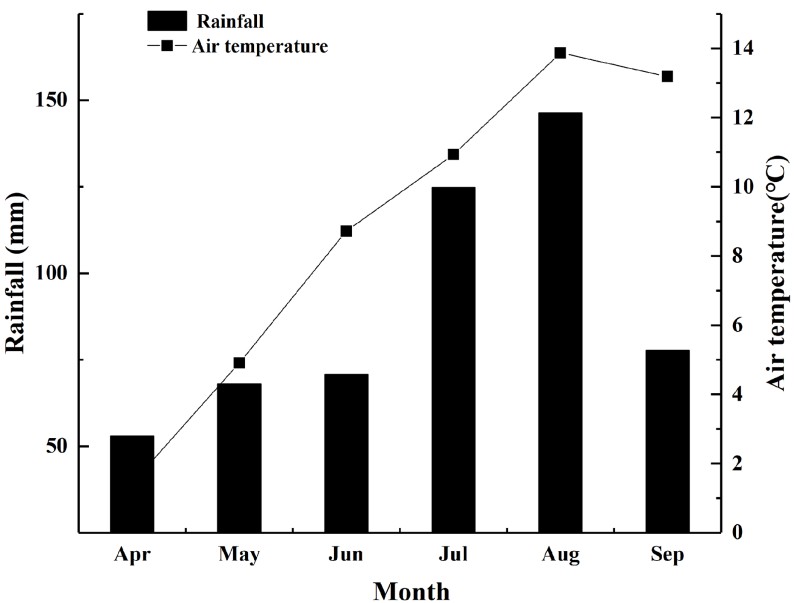

**Figure 1 Rainfall and air temperature during growing seasons in 2018.**

**Table 1 The harvest stages of triticale, rye and oat.**

| Forage species | Harvest stages (day/month) | | | | |
|---|---|---|---|---|---|
| | Heading stage | Flowering stage | Grouting stage | Milky stage | Dough stage |
| Triticale | 5/7 | 15/7 | 22/7 | 7/8 | 25/8 |
| Rye | 5/7 | 18/7 | 25/7 | 10/8 | 5/9 |
| Oat | 16/7 | 28/7 | 3/8 | 15/8 | 8/9 |

A total of 2 kg fresh forage samples were selected for each plot, and lactic acid bacteria additives Sila-Max and Sila-Mix were used at concentrations of 0.0025 and 0.5000 $g \cdot kg^{-1}$, respectively, according to the manufacturer instructions. Distilled water was used to dissolve both of additives (0.0025 or 0.5000 g Sila-Mix was dissolved in 5 mL distilled water and evenly sprayed on 1 kg fresh forage samples). An equal volume (10 ml) of distilled water was used as the control. Finally, two additives or distilled water were thoroughly mixed with the harvested forage material and then packed into laboratory silos (*i.e.*, 5 L polyethylene bottles with screw caps; Xinda Inc., Wuxi, Jiangsu, China) according to the experimental design.

The 135 silos were used to produce silage and these silos were incubated at room temperature in the laboratory for a 45-day fermentation period. Approximately 500 g samples were collected from each silo for the analysis of nutritional indices.

## Chemical analyses

To inactivate enzymes, 300 g silage samples were heated at 105 °C for 30 min. The silage samples were dried at 65 °C for 48 h to a constant weight for the DM measurements.

The total nitrogen (TN) content was measured according to the Kjeldahl method (*Krishnamoorthy et al., 1982*). The TN content was multiplied by 6.25 to calculate the CP content. The NDF and ADF contents were measured sequentially as described by *Van Soest, Robertson & Lewis (1991)*. IVDMD was determined as follows. Fresh rumen fluid was collected from rams in the morning before feeding (two generations of healthy rams were produced from the German Merino (male) × Lintao native sheep (female) hybridization). The rumen fluid samples were filtered through four layers of cheesecloth and then quickly added to Thermos bottles containing $CO_2$. The bottles were transported to the laboratory and then the samples were mixed with artificial saliva (1:2). The silage samples and the mixed fermentation broth were added to the fermenter at the same time and then $CO_2$ was added continuously for 1 min. The fermenter was immediately placed in the *in vitro* simulated incubator (ANKOM Daisy II; ANKOM Technology Inc., Macedon, NY, USA) set at 39.5 °C for the *in vitro* culture (*Menke et al., 1979*). The sample bag was a nylon cloth bag with a pore size of 40 μm. The IVDMD of the silage samples incubated in the *in vitro* simulated incubator for 48 h was measured. The relative feed value (RFV) was calculated using the following formula (*Rohweder, Barnes & Neal, 1978*):

$$RFV = \frac{DMI\ (\%BM) \times DDM\ (\%DM)}{1.29} \tag{1}$$

where DMI (DM intake) represents the free intake of roughage DM, which was calculated using formula (2), and DDM represents the digestible DM, which was predicted using formula (3).

$$DMI\ (\%BM) = \frac{120}{NDF} \tag{2}$$

$$DDM\ (\%DM) = IVDMD \tag{3}$$

## Statistical analysis

The SPSS (version 19.0) program was used to analyze the data for the DM, CP, NDF, and ADF contents and the DMY, IVDMD, and RFV. If significant differences were detected, the Duncan multiple comparison test was performed to compare the differences. Otherwise, the data were averaged and analyzed.

In an earlier study, TOPSIS (*i.e.*, multi-criteria decision-making method) was developed for a comprehensive evaluation involving a comparison between the distance $(D+i)$ between the test index and the optimal solution and the distance $(D-i)$ between the test index and the worst solution (*Hwang & Yoon, 1981*). In this study, the silage DM content, DMY, and nutritional indices resulting from different treatments underwent a TOPSIS evaluation, which was performed using DPS 7.05. The relative degree of closeness $(Ci)$ was calculated using formula (4). Increases in the $Ci$ value reflected increases in the nutritional quality of the silage.

$$C_i = \frac{D_i^-}{D_i^- + D_i^+} \tag{4}$$

For the TOPSIS comprehensive evaluation, the weights (Wi) were as follows: DMY: CP: NDF: ADF: IVDMD: RFV = 0.35: 0.20: 0.10: 0.05: 0.15:0.15.

## RESULTS

The results of the variance analysis of the nutritional quality of the silage are presented in Table 2. The extremely significant differences were detected for the single-factor effects, except for the NDF content among the additives. In terms of the effects of two factors, significant differences were not detected for the interactive effects of forage species and additives on the CP, NDF, and ADF contents, IVDMD, and RFV. Significant differences were also not detected for the interactive effects of harvest stages and additives on the CP content. However, significant differences and extremely significant differences were detected for the interactive effects of two factors on the other parameters. Extremely significant differences were detected for the interactive effects of three factors, but only for NDF and RFV. The multiple comparison test was completed for the above parameters with significant or extremely significant differences (Table 2).

### Effects of a single factor on the DMY, DM contents, and nutritional quality of silage

Forage Species: in terms of forage species, Our previous study (*Jun et al., 2022*) showed that A1, A2 and A3 were not significantly different in the average DM, but A1 was higher or significantly higher than A2 and A3 in the average DMY. The analysis of the different harvest stages and additives revealed that compared with the corresponding data for A2 and A3, the average IVDMD, and RFV were significantly higher for A1, whereas the NDF and ADF contents were significantly lower for A1 ($P < 0.05$) (Table 3). The CP contents were significantly higher for A1 than for A3. The comparison between A2 and A3 indicated the CP, NDF, and ADF contents were significantly higher for A2, but IVDMD and RFV were significantly lower for A2. Accordingly, the DMY and nutritional quality were highest for A1 (Table 3).

Harvest stages: in terms of harvest stages, as our previous study (*Jun et al., 2022*) demonstrated, as the harvest stage was delayed, the average DMY and DM content for the three forage species treated with different additives gradually increased, and both parameters were highest for B5. In contrast, the average CP content decreased significantly. Among the five harvest stages, B1 had the lowest NDF and ADF contents and the highest IVDMD and RFV. Additionally, B4 had a significantly lower NDF content than B2, B3, and B5, but its IVDMD and RFV were significantly higher than the corresponding values for B2 and B3. Therefore, the nutritional qualities were better for B1 and B4 than for the other harvest stages (Table 3).

Lactic acid bacteria additives: the additives significantly decreased the average ADF content, but had the opposite effect on the average IVDMD and RFV for the three forage

**Table 2 Variance analysis on the differences of silage nutritional quality.**

| Factor | Variable | Fermentation quality | | | | |
|---|---|---|---|---|---|---|
| | | CP | NDF | ADF | IVDMD | RFV |
| One factor | Within the forage species | 260.16** | 173.36** | 311.36** | 193.30** | 28.10** |
| | Within the harvest stages | 1313.37** | 171.98** | 260.33** | 113.84** | 41.80** |
| | Within the additives | 7.92** | 2.57 | 11.44** | 10.85** | 0.75** |
| Two factors | Forage species × Harvest stages | 32.90** | 9.61** | 26.09** | 9.46** | 83.56** |
| | Forage species × Additives | 0.2 | 1.25 | 1.62 | 0.98 | 7.22 |
| | Harvest stages × Additives | 1.94 | 2.21* | 2.93** | 2.95** | 12.07** |
| Three factors | Forage species × Harvest stages × Additives | 0.9 | 2.94** | 1.7 | 1.41 | 35.48** |

Note:
* indicates significant differences at the 0.05 level, ** indicates extremely significant differences at the 0.01 level. CP, crude protein; NDF, neutral detergent fiber; ADF, acid detergent fiber; IVDMD, *in vitro* dry matter digestibility; RFV, relative feed value.

**Table 3 Differences in the average nutrition quality for the effects of single factor (forage species, harvest stages, additives).**

| Variable | Treatment | Nutrition quality | | | | |
|---|---|---|---|---|---|---|
| | | CP (%) | NDF (%) | ADF (%) | IVDMD (%) | RFV |
| Forage species | A1 | 11.02 ± 0.30a | 52.11 ± 0.40c | 31.88 ± 0.60c | 67.89 ± 0.54a | 121.74 ± 1.76a |
| | A2 | 11.48 ± 0.40a | 58.08 ± 0.75a | 38.25 ± 0.72a | 60.25 ± 0.80c | 97.95 ± 2.69c |
| | A3 | 9.86 ± 0.31b | 53.76 ± 0.66b | 34.45 ± 0.43b | 64.62 ± 0.51b | 113.00 ± 2.26b |
| Harvest stages | B1 | 14.29 ± 0.29a | 48.30 ± 0.35d | 29.92 ± 0.24c | 70.31 ± 0.40a | 135.62 ± 1.32a |
| | B2 | 11.49 ± 0.09b | 57.82 ± 0.61a | 38.50 ± 0.71a | 60.30 ± 0.80d | 97.51 ± 2.09c |
| | B3 | 10.94 ± 0.14c | 57.95 ± 0.87a | 38.85 ± 0.57a | 61.96 ± 0.80c | 100.48 ± 2.73c |
| | B4 | 9.35 ± 0.09d | 53.72 ± 0.58c | 33.70 ± 0.88b | 64.65 ± 0.90b | 112.64 ± 2.6b |
| | B5 | 7.87 ± 0.19e | 55.45 ± 0.69b | 33.33 ± 0.67b | 64.04 ± 0.89b | 108.23 ± 2.63b |
| Lactic acid bacteria additives | C0 | 10.85 ± 0.36a | 55.08 ± 0.10a | 35.55 ± 0.70a | 63.23 ± 0.80b | 108.20 ± 2.64b |
| | C1 | 10.89 ± 0.26a | 54.41 ± 0.74a | 34.64 ± 0.34b | 64.57 ± 0.78a | 112.00 ± 2.77a |
| | C2 | 10.62 ± 0.34b | 54.45 ± 0.73a | 34.38 ± 0.70b | 64.95 ± 0.76a | 112.49 ± 2.67a |

Note:
Different letters in the same column mean significantly differences at $P < 0.05$. CP, crude protein; NDF, neutral detergent fiber; ADF, acid detergent fiber; IVDMD, *in vitro* dry matter digestibility; RFV, relative feed value.

species harvested at different stages (Table 3). The addition of Sila-Mix decreased the average silage CP content (Table 3).

## The effects of two-way interaction of harvest stages and forage species on the average DMY and nutritional quality of silage

DMY: Our previous study (*Jun et al., 2022*) showed that A1 was better than A2 and A3 in the average DMY at the B4 and B5 stage.

DM content: As our previous study (*Jun et al., 2022*) demonstrated, the DM contents of the three forage species was no significant differences at the same stage but was significant differences at different stage.

CP content: For each forage species, the average CP content was highest for the samples harvested at the B1 stage. For the additive-treated samples harvested at the same stage, the

average CP content was significantly higher for A1 and A2 than for A3. The comparison of the samples harvested at the B1 stage indicated the average CP content was significantly higher for A1 than for A2 ($P < 0.05$). Conversely, for the samples harvested at the B2 stage, the average CP content was significantly lower for A1 than for A2 ($P < 0.05$), which suggested that B1 was the best stage and also A1B1 was the best treatment for the average CP content (Table 4).

NDF and ADF contents: For each forage species, the NDF and ADF contents were low for the samples harvested at the B1 stage. For the samples harvested at the B1 stage, there were no significant differences in the average NDF and ADF contents among the three forage species, but for the samples harvested at the other stages, the average NDF and ADF contents were generally lower for A1 than for A2 and A3. Thus, it indicated that B1 was the best stage and also A1 was better than A2 and A3 for the average NDF and ADF contents (Table 4).

IVDMD and RFV: For each harvest stage, the average IVDMD and RFV were significantly higher for A1 than for A2 and A3, with the exception of IVDMD and RFV for the samples harvested at the B2 and B1 stages, respectively. For each forage species, IVDMD and RFV were highest for the samples harvested at the B1 stage, followed by the samples harvested at the B5 stage (for IVDMD) and the samples harvested at the B4 stage (for RFV), which suggested that B1 was the best harvest stage and A1 was the best forage for the average IVDMD and RFV (Table 4).

## Interactive effects of harvest stages and additives

For each harvest stage, there were no significant differences in the average NDF and ADF contents, IVDMD, and RFV among the three silages produced using different additives, which indicated that additives C1 and C2 did not significantly influence the nutritional quality of triticale, rye, and oat silages derived from samples harvested at the same stage (Table 4). For each additive and additive-free treatment, the nutritional quality was highest for the samples harvested at the B1 stage, followed by the samples harvested at the B4 stage, so B1 was the best stage compared to the other stages (Table 4).

## The effects of three-way interaction on the silage nutritional quality

The interactive effects of forage species, harvest stages, and additives significantly affected the NDF content of silage (Table 5). In terms of the forage species, for the samples harvested at the B1 stage, the NDF content for A1 was similar to that for A2, with C1 or C0, and similar to that for A3, with C1 and C2. For the other harvest stages (B2–B5), the NDF content for A1 was significantly or not significantly lower than that for A2 and A3 when the same additives were used (Table 5). Thus, A1 had the highest nutritional quality for the NDF content.

In terms of the harvest stages, the silage NDF contents for the three forage species in the same additive treatment were lowest for the samples harvested at the B1 stage, followed by the samples harvested at the B4 stage. These findings suggest that harvesting the three forage species at the B1 and B4 stages will lead to the production of silage with a low NDF content (Table 5).

**Table 4 Differences of silage nutritional quality for the effect of two-way interaction.**

| Harvest stages | Forage species/additives | CP (DM%) | NDF (DM%) | ADF (DM%) | IVDMD (DM%) | RFV |
|---|---|---|---|---|---|---|
| | A1 | 14.25 ± 0.11a | 48.34 ± 0.74gh | 29.05 ± 0.50ghi | 72.06 ± 0.34a | 138.95 ± 2.39a |
| B1 | A2 | 13.08 ± 0.09b | 49.25 ± 0.47g | 30.31 ± 0.27gh | 69.76 ± 0.41b | 131.84 ± 1.3b |
| | A3 | 12.53 ± 0.12c | 47.30 ± 0.45h | 30.41 ± 0.34g | 69.11 ± 0.82b | 136.08 ± 2.51ab |
| | A1 | 11.53 ± 0.05e | 54.74 ± 0.38d | 35.69 ± 0.70de | 63.11 ± 1.01d | 107.27 ± 1.69g |
| B2 | A2 | 11.88 ± 0.14d | 61.36 ± 0.59ab | 43.09 ± 0.49a | 55.96 ± 0.95f | 84.85 ± 1.3j |
| | A3 | 11.06 ± 0.10f | 57.37 ± 0.58c | 36.71 ± 0.46cd | 61.83 ± 0.81d | 100.42 ± 2.21h |
| | A1 | 11.26 ± 0.26ef | 53.48 ± 0.54de | 37.09 ± 0.47cd | 66.24 ± 0.58c | 115.32 ± 1.58ef |
| B3 | A2 | 11.38 ± 0.07ef | 62.41 ± 0.71a | 42.22 ± 0.46a | 57.54 ± 0.64ef | 85.83 ± 1.14j |
| | A3 | 10.17 ± 0.14g | 57.97 ± 1.26c | 37.25 ± 0.76c | 62.10 ± 0.89d | 100.3 ± 3.8h |
| | A1 | 9.42 ± 0.04h | 51.40 ± 0.39f | 28.65 ± 0.57i | 68.76 ± 0.64b | 124.51 ± 1.73c |
| B4 | A2 | 9.61 ± 0.16h | 57.44 ± 0.40c | 39.21 ± 0.36b | 59.04 ± 1.00e | 95.68 ± 1.92hi |
| | A3 | 9.02 ± 0.18i | 52.32 ± 0.61ef | 33.23 ± 0.31f | 66.14 ± 0.47c | 117.72 ± 1.57de |
| B5 | A1 | 8.66 ± 0.03j | 52.58 ± 0.56ef | 28.91 ± 0.40hi | 69.28 ± 0.56b | 122.64 ± 1.35cd |
| | A2 | 8.42 ± 0.04j | 59.93 ± 0.58b | 36.42 ± 0.55cd | 58.93 ± 0.62e | 91.56 ± 1.49i |
| | A3 | 6.52 ± 0.09k | 53.84 ± 0.35de | 34.66 ± 0.33e | 63.92 ± 0.54d | 110.48 ± 1.36fg |
| | C0 | – | 49.07 ± 0.76f | 30.00 ± 0.34d | 70.57 ± 0.54a | 133.95 ± 1.52a |
| B1 | C1 | – | 47.66 ± 0.40f | 30.06 ± 0.52d | 70.09 ± 0.91a | 136.98 ± 2.75a |
| | C2 | – | 48.16 ± 0.59f | 29.71 ± 0.43d | 70.27 ± 0.65a | 135.94 ± 2.55a |
| | C0 | – | 57.81 ± 1.24abc | 39.38 ± 1.10a | 58.86 ± 1.05d | 95.29 ± 3.48e |
| B2 | C1 | – | 58.53 ± 1.10ab | 39.07 ± 1.08a | 59.98 ± 1.16d | 95.69 ± 2.99de |
| | C2 | – | 57.13 ± 0.87abcd | 37.05 ± 1.47ab | 62.06 ± 1.76cd | 101.55 ± 4.27cde |
| | C0 | – | 59.22 ± 1.18a | 39.74 ± 0.69a | 61.29 ± 1.10cd | 96.84 ± 3.65de |
| B3 | C1 | – | 57.87 ± 1.06abc | 38.78 ± 1.17a | 61.65 ± 1.75cd | 99.75 ± 4.64cde |
| | C2 | – | 56.76 ± 2.11abcd | 38.04 ± 1.06a | 62.94 ± 1.34bcd | 104.86 ± 5.79bcde |
| | C0 | – | 53.80 ± 1.03de | 34.53 ± 1.40bc | 62.52 ± 1.68bcd | 108.84 ± 4.82bcd |
| B4 | C1 | – | 53.10 ± 1.02e | 32.67 ± 1.68cd | 66.59 ± 1.41ab | 117.26 ± 4.24b |
| | C2 | – | 54.27 ± 1.07cde | 33.90 ± 1.60bc | 64.83 ± 1.45bc | 111.81 ± 4.5bc |
| B5 | C0 | – | 55.52 ± 1.11bcde | 34.13 ± 1.16bc | 62.89 ± 1.54bcd | 106.09 ± 4.55bcde |
| | C1 | – | 54.92 ± 1.42bcde | 32.64 ± 1.07cd | 64.56 ± 1.24bc | 110.27 ± 4.61bc |
| | C2 | – | 55.93 ± 1.15bcde | 33.22 ± 1.33cd | 64.67 ± 1.88bc | 108.32 ± 4.92bcde |

Note:
Different letters in the same column mean significantly differences at *P* < 0.05. A1: triticale, A2: rye, A3: oat; B1: heading stage, B2: flowering stage, B3: grouting stage, B4: milky stage, B5: dough stage; C0: additive-free treatment, C1: Sila-Max, C2: Sila-Mix. CP, crude protein; NDF, neutral detergent fiber; ADF, acid detergent fiber; IVDMD, *in vitro* dry matter digestibility; RFV, relative feed value.

In terms of the additives, both C1 and C2 decreased the NDF contents of the A1 and A2 silages produced from the samples harvested at the B1 stage compared to C0. However, there were no significant differences in the effects of C1 and C2 on the same forage species. For the silage produced using samples harvested at the B2 stage, the addition of C2 decreased the A2 and A3 NDF contents. For the samples harvested at the B3, B4, and B5 stages, the addition of C1 decreased the silage NDF content, with the exception of the silage derived from the A2 samples harvested at the B5 stage. Moreover, C2 did not have any

**Table 5 Differences of silage nutritional quality for the effect of three-way interaction.**

| Cutting stage | Lactic acid bacteria additives | forage species | NDF (DM%) | RFV |
|---|---|---|---|---|
| B1 | C0 | A1 | 50.86 ± 0.14nop | 132.17 ± 0.57bcdef |
| | | A2 | 50.20 ± 0.47opq | 130.80 ± 2.63cdefg |
| | | A3 | 46.14 ± 0.38r | 138.87 ± 1.28abc |
| | C1 | A1 | 47.47 ± 0.77qr | 140.84 ± 3.98ab |
| | | A2 | 47.75 ± 0.36qr | 134.44 ± 2.20bcd |
| | | A3 | 47.76 ± 1.08qr | 135.68 ± 7.63abcd |
| | C2 | A1 | 49.80 ± 0.72pq | 130.27 ± 1.81cdefg |
| | | A2 | 46.70 ± 1.02r | 143.84 ± 3.97a |
| | | A3 | 47.99 ± 0.38qr | 133.70 ± 3.00bcde |
| B2 | C0 | A1 | 54.00 ± 0.59ijklm | 104.87 ± 1.63nopq |
| | | A2 | 61.89 ± 1.02abc | 82.75 ± 0.97x |
| | | A3 | 57.53 ± 1.18defgh | 98.24 ± 3.60pqrst |
| | C1 | A1 | 55.15 ± 0.43ghijk | 103.99 ± 0.97nopqr |
| | | A2 | 62.23 ± 1.21abc | 84.79 ± 2.85wx |
| | | A3 | 58.22 ± 0.47def | 98.30 ± 1.05pqrst |
| | C2 | A1 | 55.06 ± 0.87ghijk | 112.94 ± 2.50jklmn |
| | | A2 | 59.97 ± 0.48cd | 87.00 ± 2.60uvwx |
| | | A3 | 56.36 ± 1.25fghij | 104.71 ± 5.54nopq |
| B3 | C0 | A1 | 54.92 ± 0.48hijk | 110.36 ± 0.44lmno |
| | | A2 | 63.00 ± 0.12ab | 85.38 ± 0.68vwx |
| | | A3 | 59.74 ± 0.19cde | 94.79 ± 0.25rstuv |
| | C1 | A1 | 53.85 ± 0.63ijklmn | 117.00 ± 3.23ijklm |
| | | A2 | 60.05 ± 0.61cd | 87.58 ± 2.67uvwx |
| | | A3 | 59.70 ± 0.8cde | 94.68 ± 2.37rstuv |
| | C2 | A1 | 51.65 ± 0.28lmnop | 118.61 ± 0.66ijkl |
| | | A2 | 64.17 ± 1.05a | 84.55 ± 2.34wx |
| | | A3 | 54.46 ± 3.05ijkl | 111.42 ± 8.64klmn |
| B4 | C0 | A1 | 51.70 ± 1.13lmnop | 120.32 ± 3.98hijk |
| | | A2 | 57.59 ± 0.71defgh | 90.42 ± 1.13tuvwx |
| | | A3 | 52.10 ± 0.41klmnop | 115.78 ± 1.52ijklm |
| | C1 | A1 | 51.18 ± 0.46mnop | 128.58 ± 0.39defgh |
| | | A2 | 56.79 ± 0.20efghi | 101.26 ± 2.48opqrs |
| | | A3 | 51.32 ± 1.42mnop | 121.95 ± 2.43ghij |
| | C2 | A1 | 51.33 ± 0.51mnop | 124.64 ± 1.71efghi |
| | | A2 | 57.93 ± 1.00defg | 95.35 ± 2.74qrstu |
| | | A3 | 53.55 ± 1.03jklmn | 115.43 ± 2.81ijklm |

| Table 5 (continued) | | | | |
|---|---|---|---|---|
| Cutting stage | Lactic acid bacteria additives | forage species | NDF (DM%) | RFV |
| B5 | C0 | A1 | 52.56 ± 1.08klmnop | 120.56 ± 3.18hijk |
| | | A2 | 59.55 ± 0.26cde | 90.33 ± 2.21tuvwx |
| | | A3 | 54.44 ± 0.77ijkl | 107.38 ± 2.02mnop |
| | C1 | A1 | 51.44 ± 0.58lmnop | 124.41 ± 1.03efghi |
| | | A2 | 60.38 ± 0.81bcd | 93.49 ± 1.49stuvw |
| | | A3 | 52.93 ± 0.45klmno | 112.93 ± 2.66jklmn |
| | C2 | A1 | 53.75 ± 0.96jklmn | 122.95 ± 2.63fghi |
| | | A2 | 59.87 ± 1.77cd | 90.85 ± 4.06tuvwx |
| | | A3 | 54.17 ± 0.17ijklm | 111.14 ± 1.71klmn |

**Note:**
Different letters in the same column mean significantly differences at $P < 0.05$. A1: triticale, A2: rye, A3: oat; B1: heading stage, B2: flowering stage, B3: grouting stage, B4: milky stage, B5: dough stage; C0: additive-free treatment, C1: Sila-Max, C2: Sila-Mix. NDF, neutral detergent fiber; RFV, relative feed value.

uniform effects on the three silages. Accordingly, C1 decreased the NDF contents of the three forage silages to some extent (Table 5).

In terms of the forage species, for the same harvest stage and additive treatment, RFV was significantly higher for A1 than for A2, except for the samples harvested at the B1 stage. It was also significantly higher for A1 than for A3 for the samples harvested at the B3 and B5 stages. Thus, the A1 silage had the highest nutritional quality for the RFV. In terms of the additives, for the same harvest stage and forage species, the addition of C1 increased the RFV of the three silages, except for the A3 samples harvested at the B1 stage, the A1 samples harvested at the B2 stage, and the A3 samples harvested at the B3 stage. Furthermore, the addition of C2 also increased the RFV of the three silages, except for the A1 and A3 samples harvested at the B1 stage, the A2 samples harvested at the B3 stage, and the A3 samples harvested at the B4 stage. These findings suggested that C1 and C2 increased the RFV of three silages to some extent. In terms of the harvest stage, RFV was relatively high when samples were harvested at the B1 stage which indicated that B1 stage was the best for the RFV (Table 5).

## Comprehensive evaluation

The above-mentioned results implied that C1 was better than C2 for increasing the nutritional quality of the three silage types. However, DMY is a critical factor during silage production. Therefore, the DMY and nutritional quality of the silage produced using three forage species harvested at different stages and treated with C1 were comprehensively evaluated according to the TOPSIS method. The results suggested that if C1 is used as an additive, harvesting A1 and A2 at the milky stage and A3 at the dough stage will lead to the production of high-quality silage. Among the three forage species, A1 was the best for producing high-quality silage (Table 6). In conclusion, to optimize silage production on the Qinghai–Tibet Plateau, triticale variety 'Gannong No.2' should be harvested at the milky stage and Sila-Max should be added during the fermentation period.

**Table 6 TOPSIS evaluation of DMY and nutritional quality of three forages in different harvest stages for silage supplemented with C1 additive treatment.**

| Forage materials | Harvest stages | $D+$ | $D-$ | $C_i$ | Rank |
|---|---|---|---|---|---|
| A1 | B1 | 0.2100 | 0.1327 | 0.3872 | 2 |
| | B2 | 0.2318 | 0.0980 | 0.2971 | 10 |
| | B3 | 0.2162 | 0.1223 | 0.3613 | 4 |
| | B4 | 0.1617 | 0.1656 | 0.5060 | 1 |
| | B5 | 0.1096 | 0.2299 | 0.2585 | 14 |
| A2 | B1 | 0.2774 | 0.1187 | 0.2997 | 8 |
| | B2 | 0.2335 | 0.0989 | 0.2975 | 9 |
| | B3 | 0.2073 | 0.1121 | 0.3510 | 5 |
| | B4 | 0.1915 | 0.1120 | 0.3690 | 3 |
| | B5 | 0.1771 | 0.1121 | 0.1411 | 15 |
| A3 | B1 | 0.3046 | 0.1126 | 0.2699 | 12 |
| | B2 | 0.2401 | 0.0846 | 0.2605 | 13 |
| | B3 | 0.2226 | 0.0865 | 0.2798 | 11 |
| | B4 | 0.2149 | 0.1024 | 0.3227 | 7 |
| | B5 | 0.1949 | 0.1017 | 0.3429 | 6 |

Note:
A1: triticale, A2: rye, A3: oat; B1: heading stage, B2: flowering stage, B3: grouting stage, B4: milky stage, B5: dough stage.

## DISCUSSION

Our previous studies confirmed that the nutritional quality of silage varies significantly depending on the forage species, harvest stages, and additives (*Hou et al., 2017*; *Coblentz et al., 2018*), but the effects of the harvest stage and additives on the production of silage from triticale and rye plants cultivated on the Qinghai–Tibet Plateau were unknown. Clarifying how the harvest stage and additives affect the production of triticale, rye, and oat silages will provide the basis for selecting the ideal forage species, harvest stages, and additives for silage production in this region.

### Differences in the DMY of three silages and the underlying reasons on the Qinghai–Tibet Plateau

The preparation of silage not only considers its nutritional quality but also its dry matter yield (DMY). The DMY of forage in a particular unit area also affects the DMY of silage, so the DMY of forage also became one of the primary indices for evaluating silage in this experiment. The DMY of 'Gannong No. 2' triticale increased significantly from the milky stage and peaked at the dough stage, and was significantly higher than that of rye and oat in this experiment. These observations may be related to the fact that triticale has a thick stem (4.25 mm) and blade (317.50 μm) as well as large leaves (20.34 cm$^2$) and many tillers (5–6/plant) (*Liu, 2018*). Rye plants are taller than oat and triticale plants, due to smaller stems (3.21 mm) and thinner blades (243.00 μm) with smaller leaves (15.22 cm$^2$) than those of triticale. However, rye had the lowest DMY (*Dai, 2018*). The variations in rye yield and nutritive value tests carried out by *Kim et al. (2016)* in southern Oklahoma, USA, showed that the highest DMY of oat was 8.11 t/ha, and the oat silage experiment of *Zhao et al. (2018)*

on the Qinghai Tibet Plateau showed that the maximum DMY of oat for ensiling silage was 13.6–15.8 t/ha, while an experiment carried out by *Li, Tian & Du (2016)* on the evaluation of triticale varieties (lines) in Gansu Province, China, showed that the highest DMY of triticale was 16.05–16.20 t/ha. This indicated that the DMY of triticale was higher than that of rye and oat, which was consistent with the results of this experiment, but the DMY of triticale (17.84 t/ha) and rye (12.04 t/ha) in this experiment was higher than that of *Kim et al. (2016)* and *Li, Tian & Du (2016)*, which might be due to the large difference in temperature between day and night and the long sunshine hours on the Qinghai–Tibet Plateau. Temperature and sunshine affect plant nutrient accumulation; high temperatures and long sunshine are conducive to photosynthesis to produce organic matter; low temperatures at night can inhibit forage respiration and reduce the degradation of organic matter (*Zhang et al., 2015*).

## Effects of harvest stages on the nutritional quality of triticale, rye, and oat silages on the Qinghai–Tibet plateau

CP content is one of the most important indicators for evaluating the nutritional value of silage, and a high content indicates good quality (*Bolson et al., 2022*). The triticale silage test performed by *Jung et al. (2021)* in Jangsoo, Chunbuk, South Korea, showed that the CP content of triticale silage decreased from the booting stage to the heading stage, and the CP content of triticale silage at the booting stage was higher. In the present study, from the interaction of forage species and cutting stages, CP contents of triticale, rye, and oat silage at the heading stage were significantly higher than that of the flowering, grouting, milky, and dough stages, and with the delay of cutting, which decreased, and that might have resulted from the CP contents of three forage at heading stage was higher than that at other stages, and were similar to that of *Jung et al. (2021)*. The CP content of triticale, rye, and oat forage gradually decreased from the heading stage to the dough stage, which is determined by the biological characteristics of forage and has been confirmed by many scholars (*Yin et al., 2022*). However, the CP content of these three silages gradually decreased from the heading stage to the dough stage, which was due to the rapid fermentation of lactic acid bacteria to reduce the pH value of silages, inhibit the reproduction of spoilage bacteria, and reduce the loss of CP (*Fabiszewska, Zielińska & Wróbel, 2019*). At the same time, due to the large difference in the CP content of the forage itself at different growth stages, the CP content of these three silages harvested at different stages showed a gradual downward trend.

In this study, the CP content of triticale, rye, and oat treated with two lactic acid bacteria additives was not significantly higher than that of the control, except for individual treatments, which indicated that the nitrogen loss in the treatments with two additives was less during the fermentation process of silage. Although plant proteases and microorganisms can degrade the CP of silages during the fermentation process of silage and decompose it into peptides, free amino acids, ammonia, and other substances, resulting in the loss of CP (*Yuan et al., 2016*), three additives in this experiment rapidly reduced the pH value of silage fermentation, inhibiting the growth and reproduction of degrading protein microorganisms such as spoilage bacteria and clostridial bacteria, so the loss of CP was less.

The CP content of silage made from different forage varieties was different. *Harper et al. (2017)* reported that the CP content of triticale silage at the booting stage in Central County, Pennsylvania, was 17.3%, while the test of *Li et al. (2021a)* in Wuchuan, Mongolia Province, China, showed that the CP content of oat silage at the booting stage was 12.4%, indicating that the CP content of oat silage at the booting stage was lower than that of triticale. In our experiment, from the interaction of forage species and cutting stages, the CP content of triticale and rye silage was significantly higher than that of oat silage, which was determined by their genetics (*Harper et al., 2017*; *Li et al., 2021a*).

NDF is mainly composed of cellulose, hemicellulose, and lignin and ADF is a type of carbohydrate comprising pure cellulose and acid cellulose that cannot be dissolved in an acid detergent, which directly affects the DMY of forage (*Chen, Cannon & Conklin-Brittan, 2012*). From the influence of a single factor, the NDF and ADF contents of triticale silage were significantly lower than those of rye and oat, indicating that the fiber content of triticale silage was better than that of rye and oat silage. From the interaction of two factors and three factors, the NDF and ADF contents of the three forage silages depicted the same rate of change with the effect of a single factor, showing a trend of first rising and then falling and reaching a peak at the flowering or grouting stage, which was consistent with the findings of *Zhao et al. (2019)*. *Li et al. (2007)* reported that the contents of NDF and ADF in leaves and spikes of triticale are significantly higher than those in stems. From the heading stage to the dough stage, the NDF and ADF contents in stems and leaves increased significantly, while the NDF and ADF contents in spikes gradually decreased, and the fiber content decreased in spikes with grain formation. In this experiment, the three forage silages peaked at the flowering or grouting stage and then began to decline because the increase in grain maturity reduced the NDF and ADF contents of the whole plant, indicating that the grains had a greater contribution to reducing the NDF and ADF contents of the three whole plant forage silages (*Yin et al., 2022*).

## Effects of the lactic acid bacteria additives Sila-Max and Sila-Mix on the nutritional quality of triticale, rye, and oat silages

The addition of lactic acid bacteria can reduce crude fiber content, increase CP content, and improve the dry matter digestibility of silage (*Su et al., 2019*; *Li et al., 2017*). Lactic acid bacteria produced organic acid through the conversion of water-soluble carbohydrates (WSCs) into organic acids, especially lactic acid, which reduced the pH value in silage and inhibits spoilage bacteria and plant enzymes, thereby reducing the loss of CP.

The reduction in pH in silage caused by lactic acid bacteria led to acid hydrolysis of hemicellulose, the NDF content decreased, and the IVDMD increased (*Llavenil et al., 2014*). *Li et al. (2021b)* used a local lactic acid bacteria inoculant (IN1) ensile oat at the experimental base of the Sichuan Academy of Grassland Sciences on the Qinghai–Tibet Plateau and found that compared with the control group, the silage with IN1 contained a high level of CP (8.02%) and dry matter recovery (97.67%), higher dry matter digestibility (47.63 g/kg), and a low pH value. Compared with the control, the NDF, ADF, and pH values of *Leymus chinensis* silage treated with lactic acid bacteria were lower, and the WSC content was also lower (*Tian et al., 2014*). Holstein Friesian dairy cows produced more

milk, and body weight tended to be heavier when fed perennial ryegrass (*Lolium perenne*) silage fermented with lactic acid bacteria due to the feeding effect (*Ellis et al., 2016*). However, some studies have shown that lactic acid bacteria additives do not significantly increase the CP content in silage. *Nascimento Agarussi et al. (2019)* reported that compared with the control group, the addition of lactic acid bacteria did not increase the CP content of alfalfa silage; instead, it decreased the CP content during the fermentation process. *Cai et al. (2020)* reported that the CP content of corn stover and sugarcane top silage did not change notably during ensiling after adding lactic acid bacteria in Maputo, Mozambique. In the interaction of multiple factors in our experiment it was shown that although the CP content of forage treated with lactic acid bacteria additive was higher than that of the control, the effect of the lactic acid bacteria additive on the CP content of the three silages was not significant, and the results of this experiment are consistent with *Nascimento Agarussi et al. (2019)* and *Cai et al. (2020)* but are inconsistent with the results of *Li et al. (2021b)* and (*Tian et al., 2014*). This may be attributed to the fact that the CP content is increased due to the degradation of other nutrients during the silage process. Also, lactic acid bacteria additives rapidly reduced the pH value of silage, which inhibits the growth and reproduction of degrading protein microorganisms such as spoilage bacteria so that the loss of CP is less (*He et al., 2018*; *Jiang et al., 2020*).

The oat silage test of *Zhao et al. (2018)* in Qinghai–Tibet showed that only Sila-Max had a significant effect on the NDF and ADF degradation of 'Longyan No.3' oat, while Sila-Max had no significant effect on the NDF and ADF degradation of 'Longyan No.1' Also, Sila-Mix had no significant effect on the NDF and ADF degradation of both oats. Although Sila-Mix contains cobalt, which can improve the digestion of rumen fiber (*Lopez-Guisa & Satter, 1992*), these two additives did not seem to play a significant role in reducing NDF and ADF content, indicating that lactic acid bacteria do not decompose cellulose well. In the interaction of forage species × harvest stages × additives, compared to the other stages, Sila-Max and Sila-Mix significantly reduced the NDF and ADF content of each treatment at the heading stage, indicating that the cutting stage had a greater effect on the NDF and ADF content, while compared to the control, the addition of Sila-Max and Sila-Mix did not significantly reduce the NDF and ADF content at the same cutting stage (except for treatments of triticale at the heading stage). This might result from the low enzyme content and weak enzyme activity of the substrate or lactobacillus species used in this experiment, which had a low degradation effect on cellulose (*Ebrahimi et al., 2016*).

IVDMD can reflect the degradation degree of rumen microorganisms on feed in the fermentation system, and the substrate of gas production *in vitro* fermentation is mainly WCS, which can reflect the utilization of substrate by rumen microorganisms and the nutritional value of silage (*Sasson et al., 2017*). *Pérez-Pérez et al. (2019)* found that as the level of additive EM® (a cocktail of mainly lactic bacteria) increased, IVDMD increased linearly, and pH decreased, and the addition of EM® at a dose of 0.5 to 1 mL/kg DM improved IVDMD in an experiment on the effect of additive EM® on corn stover fermentation. However, *Chaturvedi et al. (2015)* reported that twelve herbal feed additives were added to a mixture of Gram straw and cowpea hay for fermentation experiments, and only three herbal feed additives (*Tephrosia purpurea*, *Ocimum sanctum* and *Emblica*

*officinalis*) significantly reduced the *in vitro* rumen methane production of silage but had no significant effect on IVDMD. In this study, in the interaction of two factors and three factors, both Sila-Max and Sila-Mix improved IVDMD in each treatment compared to the control, but the difference was not significant, except for individual treatments. It may be that substances such as cellulase in additives degrade the cellulose of silage, but the degradation effect is not significant, so the effect on increasing the content of IVDMD is not significant (*Li et al., 2019*).

## CONCLUSIONS

On the Qinghai–Tibet Plateau, the highest dry matter yield of these three forages was obtained at the dough stage. Based on the DMY, the best cutting stage of triticale and rye for producing silage was the milky stage, while the dough stage was the best cutting stage of oat for producing silage. The lactic acid bacteria additives, Sila-Max and Sila-Mix, had no significant influence on the nutritional quality of these three silages at each cutting stage. The quality of triticale silage was better than that of rye and oat after adding Sila-Max, and triticale was also the most suitable forage for producing silage among these three forages. Overall, triticale was much more economical than oat and rye in producing quality silage. The triticale variety 'Gannong No. 2' was the best forage to produce quality silage on the Qinghai–Tibet Plateau when harvested at the milky stage.

## ACKNOWLEDGEMENTS

The authors thank LiwenBianji (Edanz) and AJE for editing the English text of a draft of this manuscript.

### Funding

This work was supported by the National Natural Science Foundation (32260339), Industry Supporting Program (2022CYZC-49) and the Key Research and Development Projects (20YF8NA129) of Gansu Province, and the Major Science and Technology project of Tibet (XZ202101ZD003N), China. The funders had no role in study design, data collection and analysis, decision to publish, or preparation of the manuscript.

### Grant Disclosures

The following grant information was disclosed by the authors:
National Natural Science Foundation: 32260339.
Industry Supporting Program: 2022CYZC-49.
Key Research and Development Projects: 20YF8NA129.
Gansu Province, and Major Science and Technology Project of Tibet: XZ202101ZD003N.

### Competing Interests

The authors declare that they have no competing interests.

## Author Contributions

- Jun Ma conceived and designed the experiments, performed the experiments, analyzed the data, prepared figures and/or tables, and approved the final draft.
- Hanling Dai conceived and designed the experiments, performed the experiments, analyzed the data, prepared figures and/or tables, and approved the final draft.
- Hancheng Liu performed the experiments, analyzed the data, authored or reviewed drafts of the article, and approved the final draft.
- Wenhua Du conceived and designed the experiments, analyzed the data, authored or reviewed drafts of the article, and approved the final draft.

## Data Availability

The raw measurements are available in the Supplemental File.

## Supplemental Information

Supplemental information for this article can be found online at http://dx.doi.org/10.7717/peerj.15772#supplemental-information.

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
