# Peer review of "Effects of harvest stages and lactic acid bacteria additives on the nutritional quality of silage derived from triticale, rye, and oat on the Qinghai-Tibet Plateau"

_PeerJ, doi:10.7717/peerj.15772_

## Round 0.1 · original submission · Major Revisions

Address all the comments carefully - in particular, you must justify the reuse of previously published data.

Reviewer 1 ·

Basic reporting

The study investigated the effects of cutting stages and additives on the fermentation quality of triticale, rye and oat silage in Qinghai-Tibet Plateau. The study's findings suggest that triticale is a more suitable crop for silage production than oat or rye in the Qinghai-Tibet Plateau. It also suggests that the milky stage is the best cutting stage for triticale silage in this region. However, authors didn’t appeal that the 3 crops were cultivated in different seasons both in abstract and conclusions. Since different seasons can have different water availability, temperature, and irradiation, crop silage yield and quality in this study should not be compared without appealing their cultivation times. The study collected a large amount of data, but the results were not presented clearly. This was due to the inclusion of too many factors in the study. I have detailed suggestions in my following comments. English language is OK in this study, small typos can be found. The authors should proofread the manuscript carefully to catch any other errors.

Experimental design

1. In conclusion and results, it was not mentioned that the 3 crops were sown in different years and seasons. How can authors prove that in different seasons these crops can produce the same yield and quality? Besides, authors didn’t provide detailed climate/weather information in different seasons/months during this experiment, which is not scientifically sound.
2. Please provide timelines about how many days after sowing that different stages of different crops were harvested.
3. Line 121: What is the volume of distilled water added?
4. Line 132: Please explain IVDMD.

Validity of the findings

5. It’s advised to conclude all the interaction effects in a table (three-way and two-way) instead of just describing them in results. If the interaction effect is not significant between two factors, it’s not necessary to present each treatment combination such as those in Tables 2-5.
6. Table 1: The mean separation was done in each column within each variable. Please also check for similar problems in other tables and figures. For lactic acid bacterial additives, ADF, why the lower numbers were marked as ‘a’, higher number was marked with ‘b’?
7. Line 483: It seems lactic acid bacterial additives increase RFV and IVDMD, are these parameters reflecting nutritional quality?
8. Table notes of table 5: IVDMD instead of IVDMDF.

Reviewer 2 ·

Basic reporting

The current study investigated the effects of harvest stages and additives on the nutritional quality of triticale, rye, and oat silages and determine the optimal forage species, harvest stage, and lactic acid bacteria additive for producing high-quality silage on the Qinghai Tibet Plateau. Meanwhile, the substantial part of data (especially dry matter content and dry matter yield) included in this manuscript is against the research ethics and integrity. This data is redundant to authors previous published work, and here authors are trying to reuse substantial part of their own previously published research data. (https://doi.org/10.3390/agronomy12123113). Therefore, I recommend to reject this article with serious note.

Experimental design

No comment

Validity of the findings

Against the ethics.

Additional comments

no comment.

---

## Round 0.2 · accepted · Accept

The authors have successfully addressed all the comments.

Reviewer 1 ·

Basic reporting

I appreciate the revisions made by the authors in response to my comments. They have fully addressed all of my concerns, and I am satisfied with the manuscript. I recommend that the manuscript be accepted for publication.

Experimental design

no comment

Validity of the findings

no comment

Reviewer 2 ·

Basic reporting

Authors have addressed the raised concerns in the revised version of manuscript; hence, I recommend its publication.

Experimental design

Experimental design is fine.

Validity of the findings

Findings could be of interests for international community.

Additional comments

N/A